# A Retrospective Analysis of the American Thrombosis and Hemostasis Network Dataset Describing Outcomes of Dental Extractions in Persons with Hemophilia

**DOI:** 10.3390/jcm12051839

**Published:** 2023-02-24

**Authors:** Heather Messenger, Roshni Kulkarni, Michael Recht, Chunla He

**Affiliations:** 1Department of Pediatrics and Human Development, Michigan State University, East Lansing, MI 48824, USA; 2American Thrombosis and Hemostasis Network, Rochester, NY 14626, USA; 3Department of Pediatrics, Section of Pediatric Hematology-Oncology, Yale University School of Medicine, New Haven, CT 06520, USA

**Keywords:** ATHNdataset, dental extraction, emicizumab, epsilon aminocaproic acid, postoperative bleeding, tranexamic acid

## Abstract

Introduction: dental extractions (DEs) in persons with hemophilia A or B (PWH-A or PWH-B) are often associated with bleeding and needing hemostatic therapies (HTs). Aim: to analyze the American Thrombosis and Hemostasis Network (ATHN) dataset (ATHNdataset) to assess trends, uses and impacts of HT on bleeding outcomes following DEs. Methods: PWH seen at ATHN affiliates who underwent DEs and opted to share their data with the ATHNdataset between 2013–2019 were identified. The type of DEs, use of HT and bleeding outcomes were assessed. Results: Among 19,048 PWH ≥2 years of age, 1157 underwent 1301 episodes of DE. Those on prophylaxis experienced a nonsignificant reduction in dental bleeding episodes. Standard half-life factor concentrates were used more often than extended half-life products. PWHA were more likely to undergo DE in the first 30 years of life. Those with severe hemophilia were less likely to undergo DE than those with a mild disease (OR: 0.83; 95% CI: 0.72–0.95). PWH with inhibitors had statistically significantly increased odds of dental bleeding (OR: 2.09, 95% CI; 1.21–3.63). Conclusion: our study showed that persons with mild hemophilia and younger age were more likely to undergo DE; the presence of inhibitors increased the likelihood of bleeding, while those with prophylaxis and receiving HT experienced a non-statistically significant reduction in bleeding.

## 1. Introduction

Dental extractions (DEs) are one of the most common interventional procedures performed in persons with hemophilia A or hemophilia B (PWHA or PWHB). However, little is known about the impact of the type of hemophilia, hemostatic therapies or bleeding outcomes. Furthermore, there is no standard protocol for how to treat PWH undergoing DEs. Treatments have included pre- and postoperative CFC therapies, systemic desmopressin (DDAVP) and/or systemic and local antifibrinolytics to prevent bleeding complications. Similar overall bleeding rates have been reported to occur with pre- and postoperative factor CFC therapy compared to a single-dose preoperative CFC (11.9% and 11.4%, respectively) [1].

Our project focused on DE bleeding outcomes in PWH who received care at hemophilia treatment centers (HTCs) in the US and were American Thrombosis and Hemostasis Network (ATHN) affiliates. The latter is a national non-profit organization that partners with HTCs across the US to support research and improve the care of those affected by bleeding and clotting disorders. Participating HTCs enter local data into ATHN systems (ATHNdataset). Patients authorize the inclusion of their demographic and clinical information into this de-identified Health Insurance Portability and Accountability Act (HIPAA)-compliant dataset.

We analyzed the ATHNdataset to identify any correlation of DE outcomes with the types of hemophilia, use of hemostatic products such as plasma-derived and recombinant FVIII or IX, extended half-life factor concentrates, factor VIII mimetics, antifibrinolytics, desmopressin, inhibitor status and socio-demographic (insurance status, education level and gender) parameters.

## 2. Materials and Methods

Currently, 146 hemophilia treatment centers (HTCs) from throughout the United States are ATHN affiliates and contribute data to the ATHNdataset. Participating HTCs submit core data elements including demographics, primary diagnosis, baseline factor activity levels, inhibitor status and insurance [2].

### 2.1. Subjects

Among all participants in the ATHNdataset, patients with a primary diagnosis of hemophilia A or B, having at least one dental extraction, and ≥2 years of age at any time from 2013 to 2019, were included in this study. Subjects with a secondary diagnosis of venous thromboembolism (VTE) were excluded from the study. Observations with missing or unknown values were removed from statistical models.

### 2.2. Measures

A dental extraction bleeding episode was identified as any oral or mouth bleed which occurred during or within one week after a dental extraction procedure. Treated bleeds were defined as any oral or mouth bleeds treated with hemostatic medications during or within one week post dental extraction. This study also investigated the general bleeding treatments which occurred around the time of dental extraction that may or may not have been immediately related to DE bleeding episodes. The same strategy as for selecting dental extraction bleedings was used to identify the general on-demand bleeding treatment records (i.e., any episodic treatments during or within one week post dental extraction). General prophylactic treatments were identified as those with a regular regimen around the dental extraction episodes. Hospitalization and emergency visit events were defined as any that occurred during or one day after a dental extraction. Since dental insurance data were not actively collected in ATHNdataset, we used health insurance as a surrogate for dental insurance. Health insurance was identified as the one used around the date of a dental extraction. The severity of hemophilia was classified as severe form where the baseline factor activity level was less than 1% of normal (<0.01 IU/mL), moderate where the baseline factor activity level was 1–5% of normal (0.01–0.05 IU/mL) and mild form with the baseline factor activity between 5 and 40% (>0.05–0.40 IU/mL). Normal levels were defined as factor activity of 50–150% (0.50–1.50 IU/mL). Age at dental extraction was stratified as a categorical variable including age groups in years of 2–9.99, 10–19.99, 20–29.99, 30–39.99, 40–49.99, 50–59.99, 60–69.99 and 70–79.99, ≥80.

Data were summarized using descriptive statistics. Categorical variables were summarized using frequency and percentage. Median and interquartile range (IQR) were used to present the distribution of dosing frequency and dosing amount per Kg. Simple logistic regression models were utilized to evaluate associations between dental extraction and sociodemographic and clinical variables. Simple logistic regression models were also employed to identify factors potentially associated with dental extraction bleedings. All data manipulation and analyses were carried out in SAS version 9.4 (SAS Institute, Cary, NC, USA). A *p* value less than 0.05 was considered as being statistically significant.

## 3. Results

### 3.1. Sociodemographic and Clinical Characteristics

During the study period (2013–2019) of the 19,048 PWH in the ATHNdataset, the selection criteria yielded 1157 individuals who underwent 1301 DE episodes. Of these, 74.9% were PWHA of whom 94.4% were males. Table 1 presents the simple logistic regression modeling results of the association between dental extraction and sociodemographic and clinical characteristics among PWH.

Significant associations were observed between dental extraction and biological age at data cut (*p* < 0.001). DEs were higher among PWH aged 10–29.99 years, 40–49.99 years and 60–69.99 years than in other age groups. Figure 1 illustrates the distribution of age at dental extraction among PWH between 2013 and 2019.

Figure 1 illustrates the age distribution at DE among PWH between 2013 and 2019. It shows that hemophilia A patients are more likely to undergo DE in the first 30 years of life (about 70% of DEs performed at <30 years of age). Conversely, a higher percentage of hemophilia B patients received a DE at an older age, between 40 and 69.99 years, compared to hemophilia A patients.

The odds of dental extraction were increased by approximately 148% among male PWH compared to female PWH (OR: 2.48; 95% CI: 1.93–3.20). Since males only have a single copy of any gene located on the X chromosome, they cannot compensate for the damage to this gene with an extra copy as females can. As a result, X-related disorders such as hemophilia occur much more frequently in men [3]. Severe PWH were less likely to undergo dental extraction than mild PWH (OR: 0.83; 95% CI: 0.72–0.95). There is no documentation to indicate why people with severe hemophilia are less likely to have dental extraction than people with mild types. The odds of dental extraction did not significantly differ by race, hemophilia types or inhibitor status (*p* > 0.05). Furthermore, simple logistic regression models were performed to analyze the association of dental extraction with education and employment status among PWH aged ≥18 years at any time from 2013 to 2019. No significant associations were found between dental extraction and education or employment (*p* > 0.05). Approximately 85% of dental extractions were performed under the coverage of health insurance, and 5% were performed in those without any health insurance. PWHB were more likely to be not insured compared to PWHA (11.0% vs. 2.8%).

### 3.2. Dental Extraction Clinical Outcomes and Related Treatments

Figure 2 shows the number of dental extractions among PWH who underwent at least one dental extraction from the year 2013 to 2019.

A majority (90%) of PWH had only one dental extraction between 2013–2019. PWHA patients were more likely to receive two or more dental extractions compared to PWHB patients. Overall, only a small portion N = 66/1301 (approximately 5%) of patients who underwent dental extractions reported oral/mouth bleeds. Of these, 28.0% PWHA and 12.5% PWHB patients reported two or more oral/mouth bleeds during and one week post DE. Overall, approximately 18% of PWH received on-demand treatment between 2013 and 2019 during or within one week post dental extraction. Among oral-mouth bleeds related to dental extraction, 34.0% and 50.0% of PWHA and PWHB, respectively, were treated with hemostatic medications. Around the time of dental extraction, 7.4% reported hospitalization and 1.3% had a record of an emergency visit among PWHA patients, while 8.8% of hospitalization and 0.9% of emergency visits were identified among PWHB. Other hemostasis medications, Amicar was the commonly used episodic therapy, followed by standard half-life (SHL) products. Prophylactic treatment was more prevalent among patients with hemophilia A compared to the ones with hemophilia B (37.3% vs. 25.2%). SHL was the most popular prophylactic regimen for both hemophilia A and B patients undergoing dental extraction, followed by EHL products. Median and interquartile range (IQR) of dosing values and dosing frequency are also summarized in Table 2.

About half of the PWHA did not use any therapies to prevent or manage bleeding events related to dental extraction. A higher percentage (60%) of PWHB were not prescribed any medications around the dates of dental extractions. The data were not stratified by level of severity which could be why there is no significant difference between patients who received therapy and those who did not. PWH’s decision to use therapies is voluntary, so it would be interesting to know the number of treatments where an authority decision was made to determine any correlation.

### 3.3. Factors Associated with Dental Extraction Bleeding

Simple logistic regression models were applied to identify factors that might be associated with dental bleeding episodes among PWH. People with dental extractions performed using a prophylactic regimen were found to experience a nonsignificant reduction in dental bleeding episodes. Similarly, using EHL products tended to nonsignificantly reduce the risk of dental bleeding. No significant associations of dental extraction bleedings were observed with age at dental extraction, education, employment status or severity levels either. PWH who have ever developed an inhibitor significantly increased the odds of dental bleeding (OR: 2.09, 95% CI: 1.21–3.63). However, no significant relationship was found between inhibitor and dental extraction bleedings among PWH who were ever exposed to Emicizumab (*p* = 0.168). The authors caution these results as the severity level was not specified which could be why there was no significant difference between patients who received therapy and those who did not.

## 4. Discussion

This study allowed us to determine the patterns in prophylactic factor use and its impact on bleeding outcomes in 1157 PWH, over the age of 2 years, who underwent 1301 episodes of DEs and were seen at the US ATHNaffiliate sites between the years 2013 and 2019. DEs resulting in bleeding in PWH have been reported to range from 2% to 38% [1,3,4,5,6,7]. However, no consensus exists as to the optimal protocol for the prevention of bleeding associated with DEs in this population. We hypothesized that PWH whose primary hematological regimen was prophylaxis would experience fewer bleeding episodes than those who were treated episodically. In addition, we hypothesized that the use of extended half-life products would result in fewer bleeding episodes than the use of standard half-life (SHL) and nonfactor products.

In our retrospective analysis of the ATHNdataset, PWH with a severe disease were less likely to undergo DEs than those with a mild or moderate disease. DE clinical outcomes and related treatments indicated that approximately 34% of PWH used prophylaxis versus on-demand (18%) therapy for dental/mouth bleeding. Approximately 5% of PWH who underwent DEs reported oral/mouth bleeds within one week of the procedure. There was no statistically significant difference in dental bleeding episodes between those who received prophylactic factor infusion and those who were treated for dental bleeding. Similarly, no statistically significant differences in bleeding rate following DE were found in those using extended half-life factor (EHL) compared to standard half-life products (SHL). Prophylaxis was more prevalent among PWH-A compared to hemophilia B (37.3% vs. 25.2%, respectively). SHL products were more commonly used than EHL products for on-demand treatment (8.3% vs. 0.5%, respectively) as well as prophylaxis (19.8% vs. 8.1%). PWH who developed an inhibitor had significantly increased odds of dental bleeding unless they were on Emicizumab. Although more hospitalizations were reported as compared to emergency department (ED) visits around the date when DEs were performed, these numbers should be interpreted with caution because events of hospitalization or emergency visits may have been caused by reasons other than dental extractions. It would be useful to know how many hospitalizations or emergency visits were made for complications of DEs, if data are available, perhaps by requesting the data directly from the HTC affiliates of the ATHN. In the future, it would be interesting for the NHA dataset to collect systematized data to address outstanding issues.

The World Federation of Hemophilia (WFH) has published guidelines on FVIII/FIX dosing for dental procedures [4]. In addition, these guidelines advise the use of antifibrinolytic agents (tranexamic acid (TXA) and/or epsilon aminocaproic acid (EACA)) and local hemostatic measures to reduce the need for clotting factor concentrate (CFC) replacement therapy to prevent hemorrhagic complications from being sufficient for most single extractions [5].

Previous reports have also demonstrated the efficacy of TXA and EACA in reducing bleeding as well as reducing the use of CFC following dental procedures. Van Galen et al. [6] reviewed two trials of the effectiveness of local or systemic antifibrinolytics in PWH in preventing bleeding complications and either replacing or reducing the need for CFC [6]. Both trials showed a reduction in the number of bleeds after DEs, in the amount of blood loss and in the need for CFC in PWH treated with antifibrinolytics compared to those who received a placebo. The number of participants needing postoperative CFC therapy was significantly higher in the placebo group (*p* = 0.02).

Champagne et al. [7] compared the outcomes and use of hemostatic treatment and pre-and post-implementation of a standardized protocol for dental procedures in persons with inherited bleeding disorders at a single HTC. The protocol for hemostatic treatment was based on the invasiveness of the dental procedure and the proposed anesthesia. A total of 17 procedural bleedings were reported (12.4%) in 95 patients, of which 14.8% and 8.9% occurred in the control and intervention groups, respectively. No major bleeding occurred. Tranexamic acid was used more consistently after protocol implementation (72.8% vs. 89.3%, *p* = 0.019), while factor concentrate use decreased (65.4% vs. 44.6%, *p* = 0.016) and desmopressin use remained constant (46.4% vs. 32.1%, *p* = 0.100). The authors concluded that the use of a standardized protocol increased the use of TXA with a non-statistically significant reduction in procedural bleeding.

Although the HAVEN trials were not intended to study the outcomes of surgeries in people treated with Emicizumab, 64 dental procedures were reported and they showed that no preventive clotting factor was used in 42 dental procedures: 29 out of 42 had no bleeds after surgery and 13 out of 42 (30.9%) had bleeds that occurred after surgery. Nine (21% of) bleeds were treated with clotting factor and four (10% of) bleeds were not treated; five bleeds occurred after surgery and were treated with clotting factor [8].

Pell et al. [9] state that TXA is reportedly superior to EACA, is less toxic, more potent weight for weight and has a longer half-life, thus necessitating less frequent administrations of the drug [9]. The only FDA-approved usage for oral TXA is for heavy menstrual bleeding and short-term prevention in patients with hemophilia; this includes tooth extractions in patients with hemophilia as well as menorrhagia in these patients. TXA is also available as an injectable solution and indicated for short-term use (2–8 days) in PWH to reduce the risk of hemorrhage during and following tooth extraction. It is not currently indicated for local application and may not be covered under insurance. Providers may be required to obtain prior authorization for TXA or prescribe Amicar.

Our overarching hypothesis was that PWH treated prophylactically would experience less bleeding than those treated on demand. Our findings did not show a significant difference between prophylaxis and on-demand treatment in bleeding outcomes. Our retrospective results showed a low bleeding rate during and one week post DE of approximately 5% compared to reported rates in the literature of 1.9–37.5% [1,7,10,11,12,13,14,15]. In addition, our study, however, did find that the use of any treatment for bleeding, such as Amicar, factor therapies, etc., was associated with less bleeding.

### 4.1. Further Research

The significance of the local topical use of antifibrinolytic agents, such as TXA or EACA, either applied directly in the wound, via gauze saturated with TXA or EACA, or in the form of mouthwash solution in the prevention of postoperative bleeding, has been emphasized by many authors [10,16,17,18,19,20,21,22]. Two studies showed that a combination of both systemic and local antifibrinolytic use gives favorable results [18,22]. Our results along with the findings of Van Galen et al. [6] suggest that additional data collection is needed to clarify the role that antifibrinolytic agents play, including the type of antifibrinolytic (epsilon aminocaproic acid and tranexamic acid) route of delivery (systemic and topical) and form (liquid, paste and popsicle) with regard to the impact on bleeding outcomes following DEs [6].

In the future, ATHN could collect data to establish comparisons between conventional non-surgical techniques and the use of a variety of surgical procedures, including ED, that could lead to a higher risk of bleeding during or after surgery.

### 4.2. Limitations

The major limitation of our study is the limited data that currently exists on PWH with DEs in the ATHNdataset. This research did not stratify outcomes by type of DE procedure risk such as simple or surgical. Details about hospitalizations and ED visits were not captured.

In the US, the ATHNdataset provides a platform for the collection of data specific to persons diagnosed with either bleeding or clotting disorder. The core data elements, which should be collected and properly updated on every patient entered in the ATHNdataset, are voluntarily entered by HTCs. The limitations of the ATHNdataset include missing data elements, potential non-representation of the entire hemophilia population and possible data entry from sites that have resources such as a dedicated data entry person. ATHN, however, continues its commitment to support ATHN affiliates to enhance data management capacity and data quality through a program termed ATHN Data Quality Counts. The collected data are provided voluntarily instead of being systemized which could be why there is no significant difference between patients who received therapy and those who did not.

## 5. Conclusions

In this study, it was observed that while DEs tended to be performed in persons with mild hemophilia, the use of hemostatic therapies and factor prophylaxis reduced bleeding. Moreover, to prevent post-DE bleeding, the successful dental management of patients with hereditary bleeding disorders requires close collaboration between hematology teams and dentists. The ATHN*dataset allowed studies of outcomes in a large cohort of hemophilia patients.

## Figures and Tables

**Figure 1 jcm-12-01839-f001:**
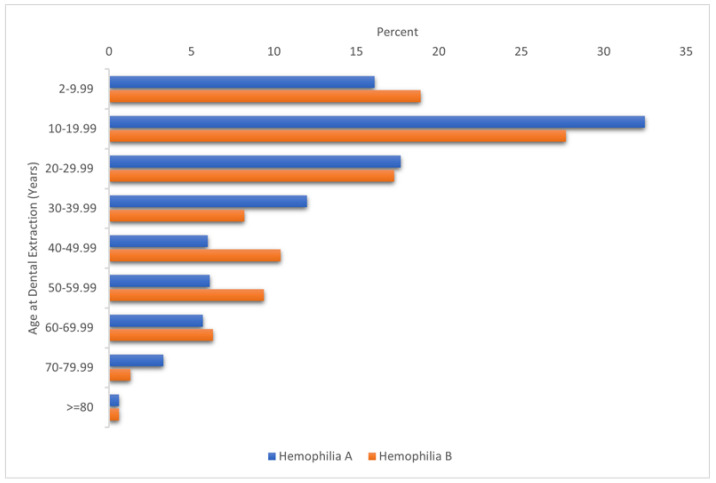
The distribution of age at dental extraction among hemophilia patients, 2013–2019.

**Figure 2 jcm-12-01839-f002:**
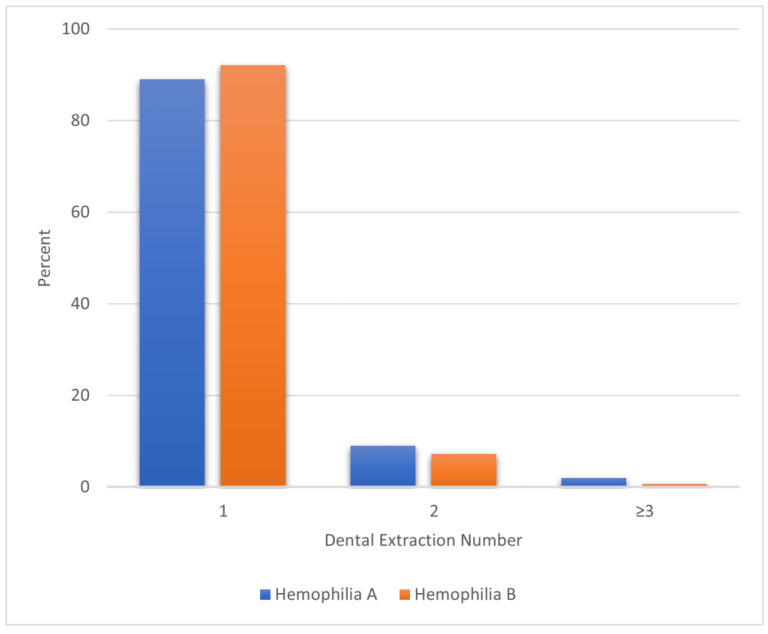
Number of dental extractions by hemophilia type, 2013–2019.

**Table 1 jcm-12-01839-t001:** Sociodemographic and clinical characteristics of hemophilia patients with and without dental extraction, and overall hemophilia patients, 2013–2019.

Variable	Dental Extraction (%)	No Dental Extraction (%)	OR (95% CI)
Most recent biological age			
2–9.99 years	103 (8.9)	2542 (14.2)	Reference
10–19.99 years	311 (26.9)	4230 (23.6)	1.81 (1.44–2.28)
20–29.99 years	324 (28.0)	3610 (20.2)	2.22 (1.76–2.78)
30–39.99 years	137 (11.8)	2802 (15.7)	1.21 (0.92–1.57)
40–49.99 years	89 (7.7)	1602 (9.0)	1.37 (1.03–1.83)
50–59.99 years	73 (6.3)	1365 (7.6)	1.32 (0.97–1.79)
60–69.99 years	72 (6.2)	1006 (5.6)	1.77 (1.30–2.41)
70–79.99 years	35 (3.0)	550 (3.1)	1.57 (1.06–2.33)
≥80 years	13 (1.1)	184 (1.0)	1.74 (0.96–3.16)
Gender			
Female	65 (5.6)	2302 (12.9)	Reference
Male	1092 (94.4)	15,588 (87.1)	2.48 (1.93–3.20)
Race ^a^			
White	955 (84.1)	14,421 (82.8)	Reference
Black or African American	118 (10.4)	1870 (10.7)	0.95 (0.78–1.16)
Asian	43 (3.8)	634 (3.6)	1.02 (0.75–1.40)
Other races	20 (1.8)	488 (2.8)	0.62 (0.39–0.97)
Diagnosis			
Hemophilia A	866 (74.9)	13,797 (77.1)	Reference
Hemophilia B	291 (25.2)	4094 (22.9)	1.13 (0.99–1.30)
Severity ^a^			
Mild	386 (34.5)	5146 (31.6)	Reference
Moderate	263 (23.5)	3582 (22.0)	0.98 (0.83–1.15)
Severe	470 (42.0)	7572 (46.5)	0.83 (0.72–0.95)
Ever inhibitor ^a^			
No	920 (85.0)	11,891 (86.0)	Reference
Yes	162 (15.0)	1934 (14.0)	1.08 (0.91–1.29)
Education ^b^			
Advanced degree	44 (9.6)	724 (10.8)	Reference
College	67 (14.7)	1072 (16.0)	1.03 (0.70–1.52)
Some college	155 (33.9)	2097 (31.2)	1.22 (0.86–1.72)
GED or equivalent	32 (7.0)	441 (6.6)	1.19 (0.75–1.91)
Secondary or under	159 (34.8)	2381 (35.5)	1.10 (0.78–1.55)
Employment ^b^			
Full-time	258 (51.8)	4089 (52.9)	Reference
Part-time	52 (10.4)	631 (8.2)	1.31 (0.96–1.78)
Not employed	188 (27.8)	3013 (39.0)	0.99 (0.82–1.20)

Abbreviations: OR: odds ratio; CI: confidence interval. ^a^. Missing values were removed from the analysis. ^b^. Only restricted to a subset of subjects aged ≥18 years at any time between 2013 and 2019, and without missing values.

**Table 2 jcm-12-01839-t002:** Clinical outcomes and treatments related to dental extraction episodes among hemophilia patients, 2013–2019 (*n* = 1301 dental extraction episodes).

Variable	Hemophilia A (*n* = 983)	Hemophilia B (*n* = 318)	All Patients (*n* = 1301)
Dental-extraction-related bleeds, *n* (%)	50 (5.1)	16 (5.0)	66 (5.1)
≥2 dental extractions	14 (28.0)	2 (12.5)	16 (24.2)
Treated dental-extraction-related bleeds ^a^	17 (34.0)	8 (50.0)	25 (37.8)
Hospitalization ^b^, *n* (%)	74 (7.4)	28 (8.8)	102 (7.8)
Emergency visit ^b^, *n* (%)	13 (1.3)	3 (0.9)	16 (1.2)
Health insurance, *n* (%)			
Insured	849 (86.4)	251 (78.9)	1100 (84.6)
Uninsured	27 (2.8)	35 (11.0)	62 (4.8)
Unknown	107 (10.2)	32 (10.1)	139 (10.7)
General bleeding treatment one week post dental extraction ^c^			
On-demand ^d^, *n* (%)	182 (18.5)	58 (18.2)	240 (18.4)
EHL products	5 (0.5)	2 (0.6)	7 (0.5)
SHL products	82 (8.3)	26 (8.2)	108 (8.3)
Plasma-derived products	7 (0.7)	3 (0.9)	10 (0.8)
Bypassing agents	5 (0.5)	2 (0.6)	7 (0.5)
Other hemostasis medications	109 (11.1)	33 (10.4)	142 (10.9)
Prophylaxis	367 (37.3)	80 (25.2)	447 (34.3)
EHL products, *n* (%)	71 (7.2)	35 (11.0)	106 (8.1)
Dose per Kg (IU/Kg), median (IQR)	50 (42–60)	76 (53–100)	51 (50–70)
Dose frequency, median (IQR)	3.5 (3.5–4.0)	7.0 (7.0–7.0)	3.5 (3.5–7.0)
SHL products, *n* (%)	221 (22.5)	36 (11.3)	257 (19.8)
Dose per Kg (IU/Kg), median (IQR)	40 (32–49)	66 (55–76)	40 (35–50)
Dose frequency, median (IQR)	2.3 (2.0–3.5)	3.5 (2.3–3.5)	2.3 (2.0–3.5)
Plasma-derived products, *n* (%)	28 (2.8)	6 (1.9%)	34 (2.6)
Dose per Kg (IU/Kg), median (IQR)	40 (31–85)	75 (74–90)	74 (38–89)
Dose frequency, median (IQR)	2.3 (2.0–2.3)	3.5 (3.5–7.0)	2.3 (2.0–3.5)
Bypassing agents, *n* (%)	10 (1.0)	2 (0.6)	12 (0.9)
Non-factor molecules, *n* (%)	24 (2.4)	NA	24 (1.8)
Other hemostasis medications, *n* (%)	2 (0.2)	1 (0.3)	3 (0.2)
Dental extraction without hemostatic treatment, *n* (%)	496 (50.5)	190 (59.7)	686 (52.7)

Abbreviations: EHL, extended half-life; SHL, standard half-life. IQR, interquartile range. ^a^. Treated oral or mouth bleeds occurred on the date of or within one week of post dental extraction episode. ^b^. Any hospitalization or emergency visits on the date of or one day after a dental extraction episode. ^c^. Any hemostatic treatments which may or may not be immediately triggered by a dental extraction bleed. ^d^. Any episodic hemostatic treatments that occurred during or within one week of post dental extraction episode.

## Data Availability

The data that support the findings of this study originate from the ATHNdataset and are available from the American Thrombosis and Hemostasis Network (ATHN). Restrictions apply to the availability of these data, which were used under license for this study. Data inquires can be made by emailing ATHN at support@athn.org.

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
