# Peer review of "A Retrospective Analysis of the American Thrombosis and Hemostasis Network Dataset Describing Outcomes of Dental Extractions in Persons with Hemophilia"

_jcm, 2023, doi:10.3390/jcm12051839_

Round 1
Reviewer 1 Report
Excessive bleeding after dental procedures is one of the most frequent complications occurring in patients with hemophilia. The study, with a large number of the sample, aims to evaluate the impact of the type of hemophilia, hemostatic therapies and bleeding outcomes. The data reported are clear and interesting even if sometimes incomplete because they are not systematized but provided on a voluntary basis.
1. Section 3.2 and 3.3 In this study only a small portion of patients (5%) who underwent dental extractions reported oral/ mouth bleeds despite a high percentage of patients (50-60%) did not use any therapies to prevent or manage bleeding events related to DEs. Is it possible that the high proportion of PWH who did not bleed after DEs was related to whether most patients were mild or moderate? If so, it would be useful to know the distribution by severity in patients whose DEs was not complicated by bleeding. This could be the reason why in the study there is no significant difference between patients who received therapy and those who did not.
2. Discussion line 197-200 Athough more hospitalizations were reported as compared to ED visits around the date when DEs were performed, the authors themselves report that these numbers should be interpreted with caution because events of hospitalization or emergency visit may have been caused by reasons other than dental extraction. Thu it would be useful to know how many hospitalizations or emergency visits were made for complications of Des, if data are available, perhaps by requesting the data directly from the HTC affiliates of the ATHN.
3. Discussion from line 201 It would be interesting to know how many people have used antifibrinolytic agents, in the light of the latest WFH guidelines and the authors' own considerations, if data are available
It would be interesting if in the future the ATHNdataset collected systematized data, in order to provide answers to the outstanding questions.
Author Response
- Comments and Suggestions for Authors
Excessive bleeding after dental procedures is one of the most frequent complications occurring in patients with hemophilia. The study, with a large number of the sample, aims to evaluate the impact of the type of hemophilia, hemostatic therapies and bleeding outcomes. The data reported are clear and interesting even if sometimes incomplete because they are not systematized but provided on a voluntary basis.
The collected data is provided voluntarily instead of systemized which could be why there is no significant difference between patients who receive therapy and those who did not.
- Section 3.2 and 3.3 In this study only a small portion of patients (5%) who underwent dental extractions reported oral/ mouth bleeds despite a high percentage of patients (50-60%) did not use any therapies to prevent or manage bleeding events related to DEs. Is it possible that the high proportion of PWH who did not bleed after DEs was related to whether most patients were mild or moderate? If so, it would be useful to know the distribution by severity in patients whose DEs was not complicated by bleeding. This could be why there is no significant difference between patients who received therapy and those who did not.
The data were not stratified by level of severity which could be why there is no significant difference between patients who received therapy and those who did not.
- Discussion line 197-200 Athough more hospitalizations were reported as compared to ED visits around the date when DEs were performed, the authors themselves report that these numbers should be interpreted with caution because events of hospitalization or emergency visit may have been caused by reasons other than dental extraction. Thus it would be useful to know how many hospitalizations or emergency visits were made for complications of Des, if data are available, perhaps by requesting the data directly from the HTC affiliates of the ATHN.
It would be useful to know how many hospitalizations or emergency visits were made for complications of Des, if data are available, perhaps by requesting the data directly from the HTC affiliates of the ATHN.
In the future, it would be interesting for the NHA data set to collect systematized data to address outstanding issues.
Reviewer 2 Report
The article submitted presents interesting results, although there are some shortcomings:
The authors discussed the topic correctly, presented appropriate previously published studies correctly stressed.
Abbreviation used for the first time in text, with no detail upon the term (CFC therapy, which only from the context could be considered clotting factor concentrate, as, otherwise, is used as abbreviation for other meanings)
The same for ED
In Chapter 2.2. is to correct month bleeds.
The following observation need to be more commented, with discussions upon other literature data or involvement of gender differences, in the Discussion section: The odds of dental extraction were increased by approximately 148% among male PWH compared to female PWH.
The same for the claimed result, especially in comparison to the literature data: Severe PWH were less likely to undertake dental extraction than mild PWH.
It would be of interest to make comparisons between nonsurgical conventional techniques and the use of all kinds of surgical procedures including DE that could trigger a higher risk of bleeding either during or post-surgery.
In chapter 3.2, the authors claim: About a half of PWHA did not use any therapies to prevent or manage bleeding events related to dental extraction. A higher percentage (60%) of PWHB were not prescribed any medications around the dates of dental extractions. In order to make comparisons from the prophylactic management point of view, it would be more accurate to refer as for the same point of view, not for some of them being considered a voluntarely decision and for the other group, an authority decision.
Author Response
The article submitted presents interesting results, although there are some shortcomings:
The authors discussed the topic correctly, presented appropriate previously published studies correctly stressed.
Abbreviation used for the first time in text, with no detail upon the term (CFC therapy, which only from the context could be considered clotting factor concentrate, as, otherwise, is used as abbreviation for other meanings.
The same for ED.
In Chapter 2.2. is to correct month bleeds.
The following observation need to be more commented, with discussions upon other literature data or involvement of gender differences, in the Discussion section: The odds of dental extraction were increased by approximately 148% among male PWH compared to female PWH.
Since males only have a single copy of any gene located on the X chromosome, they cannot compensate for the damage to this gene with an extra copy as females can. As a result, X-related disorders such as hemophilia occur much more frequently in men. (citation needed).
The same for the claimed result, especially in comparison to the literature data: Severe PWH were less likely to undertake dental extraction than mild PWH.
There is no documentation to indicate why people with severe hemophilia are less likely to have dental extraction than people with mild types.
It would be of interest to making comparisons between nonsurgical conventional techniques and the use of all kinds of surgical procedures including DE that could trigger a higher risk of bleeding either during or post-surgery.
In the future, ATHN could collect data to Establish comparisons between conventional non-surgical techniques and the use of a variety of surgical procedures, including ED, that could lead to a higher risk of bleeding during or after surgery.
In chapter 3.2, the authors claim: About a half of PWHA did not use any therapies to prevent or manage bleeding events related to dental extraction. A higher percentage (60%) of PWHB were not prescribed any medications around the dates of dental extractions. In order to make comparisons from the prophylactic management point of view, it would be more accurate to refer as for the same point of view, not for some of them being considered a voluntarely decision and for the other group, an authority decision.
PWH's decision to use therapies is voluntary, so it would be interesting to know the number of treatments where an authority decision has been made to determine any correlation.
Reviewer 3 Report
The manuscript provides a good overview of operations, especially tooth extractions, in patients withhemophilia A or B with different degrees of severity and different therapy options.
I also think it can be a good guide for doctors who care for patients with hemophilia A or B who are due for a tooth extraction. The results underscore the expected outcomes.
Author Response
Thank you for your comments.